# Vulnerability-Aware Instance Reweighting For Adversarial Training

**Olukorede Fakorede***
*Department of Computer Science*
*Iowa State University*
*fakorede@iastate.edu*

**Ashutosh Nirala**
*Department of Computer Science*
*Iowa State University*
*aknirala@iastate.edu*

**Modeste Atsague**
*Department of Computer Science*
*Iowa State University*
*modeste@iastate.edu*

**Jin Tian**
*Department of Computer Science*
*Iowa State University*
*jtian@iastate.edu*

**Reviewed on OpenReview:** *https://openreview.net/forum?id=kdPcLdJbt1*

## Abstract

Adversarial Training (AT) has been found to substantially improve the robustness of deep learning classifiers against adversarial attacks. AT involves obtaining robustness by including adversarial examples in training a classifier. Most variants of AT algorithms treat every training example equally. However, recent works have shown that better performance is achievable by treating them unequally. In addition, it has been observed that AT exerts an uneven influence on different classes in a training set and unfairly hurts examples corresponding to classes that are inherently harder to classify. Consequently, various reweighting schemes have been proposed that assign unequal weights to robust losses of individual examples in a training set. In this work, we propose a novel instance-wise reweighting scheme. It considers the vulnerability of each natural example and the resulting information loss on its adversarial counterpart occasioned by adversarial attacks. Through extensive experiments, we show that our proposed method significantly improves over existing reweighting schemes, especially against strong white and black-box attacks.

## 1 Introduction

The practical application of deep learning in safety-critical domains has been thrown into doubt following the observed brittleness of deep learning to well-crafted adversarial perturbations (Szegedy et al., 2013). This chilling observation has led to an array of methods aimed at making deep learning classifiers robust to these adversarial perturbations. Prominent among these proposed methods is adversarial training (Goodfellow et al., 2015; Madry et al., 2018). Adversarial training (AT) is an effective method that typically involves the introduction of adversarial examples in training a deep learning classifier. Several AT variants have been proposed yielding modest improvements (Wang et al., 2019; Ding et al., 2019; Kannan et al., 2018; Zhang et al., 2019). More recently, methods have been proposed to boost the performance of the existing AT variants even further, including adversarial weight perturbation (Wu et al., 2020), utilizing hypersphere

embedding (Pang et al., 2020; Fakorede et al., 2023), and augmenting dataset with unlabeled and/or extra labeled data (Carmon et al., 2019; Alayrac et al., 2019; Zhai et al., 2019).

Despite the generally impressive performance of AT against adversarial attacks, Xu et al. (2021) raised concerns about fairness in AT. It is observed that AT encourages significant disparity in natural and robust accuracy among different classes. Furthermore, AT disproportionately hurts the robust accuracy of input examples that are intrinsically harder to classify by a naturally trained classifier. For example, a naturally trained PreactResNet-18 on the CIFAR-10 dataset classifies "cat" and "ship" at approximately 89% and 96% accuracy, respectively, while the robust accuracy of "cat" and "ship" produced by an adversarially trained PreactResNet-18 on PGD-attacked CIFAR-10 dataset are approximately 17% and 59% respectively (Xu et al., 2021). In other words, the gap between the standard accuracy and robust accuracy for "cat" is 72%, whereas the difference between the standard and robust accuracy for "ship" is 37%. This disparity suggests that adversarial examples corresponding to certain classes may be treated differently by AT.

Adversarial examples are crafted from natural examples that exhibit varying degrees of intrinsic vulnerability measured by their closeness to the class decision boundaries. Intuitively, adversarial examples corresponding to intrinsically vulnerable examples are moved farther across the decision boundary into wrong classes which makes them easy to misclassify. Zhang et al. (2020) observed that AT encourages learning adversarial variants of less vulnerable natural examples at the expense of the intrinsically susceptible ones as the training proceeds. This may partly explain the phenomenon of robust overfitting (Chen et al., 2020; Zhang et al., 2020). Therefore, the performance of AT may be improved by assigning higher weights to the robust losses of adversarial variants of vulnerable natural examples.

The idea of reweighting robust losses of adversarial examples has recently been explored in the literature. *GAIRAT* (Zhang et al., 2020) assigns smaller or larger weights to the robust losses of adversarial examples based on the geometric distance of their corresponding natural counterpart to the decision boundary. Specifically, *GAIRAT* measures the geometric distance using the least number of PGD steps needed to misclassify the natural example. As a result, the *GAIRAT* reweighting function can only take a few values (corresponding to discrete PGD steps) and is unstable because its value depends largely on the initial starting point of each natural example which may change depending on the attack path (Liu et al., 2021). Liu et al. (2021) proposed reweighting robust losses based on the margin between the estimated ground-truth probability of the adversarial example and the probability of the most confusing label. We contend that this method does not exploit information about the intrinsic vulnerability of a natural example. More importantly, we observe that these methods only yield competitive robustness on attacks like PGD and FGSM, but underperform against stronger white-box or black-box attacks CW, AA, or SPSA.

This paper proposes a novel instance-wise weight assignment function for assigning importance to the robust losses of adversarial examples used for adversarial training. Our weight assignment function considers the intrinsic vulnerability of individual natural examples from which adversarial examples used during training are crafted. We capture the intrinsic vulnerability of each natural example using the likelihood of it being correctly classified, which we estimate using the model's confidence about the example belonging to its true class. In addition, we argue that adversarial attacks have a unique impact on each individual example. This contributes to the disparity in robustness accuracy exhibited by examples of different classes. Hence, we compute the discrepancies between a model's prediction on each natural example and its corresponding adversarial example as another measure for the vulnerability of the example.

We summarize the contributions of this paper as follows:

1. We propose a novel Vulnerability-aware Instance Reweighting ( *VIR*) function for adversarial training. The proposed reweighting function takes consideration of the intrinsic vulnerability of individual examples used for adversarial training and the information loss occasioned by adversarial attacks on natural examples.

2. We experimentally demonstrate the effectiveness of the proposed reweighting strategy in improving adversarial training.

3. We show that existing reweighting methods *GAIRAT* (Zhang et al., 2020) and *MAIL*(Liu et al., 2021) only yield significant robustness against attacks FGSM and PGD at the expense of stronger

attacks CW (Carlini & Wagner, 2017), Autoattack(Croce & Hein, 2020b), and FMN (Pintor et al., 2021). Using various datasets and models, we show that the proposed *VIR* method consistently improves over the existing reweighting methods across various white-box and black-box attacks.

## 2 RELATED WORK

**Adversarial Attacks.** Since it became known that deep neural networks (DNN) are vulnerable to norm-bounded adversarial attacks (Biggio et al., 2013; Szegedy et al., 2013), a number of sophisticated adversarial attack algorithms have been proposed (Goodfellow et al., 2015; Madry et al., 2018; Carlini & Wagner, 2017; Athalye et al., 2018; Dong et al., 2018; Moosavi-Dezfooli et al., 2016). Adversarial attacks can broadly be classified into white-box and black-box attacks. White-box attacks are crafted with the attacker having full access to the model parameters. Prominent white-box attacks include Fast Gradient Sign Method (Goodfellow et al., 2015), DeepFool (Moosavi-Dezfooli et al., 2016), C&W (Carlini & Wagner, 2017), Projected Gradient Descent (PGD) (Madry et al., 2018) etc. On the other hand, in black-box settings, the attacker has no direct access to the model parameters, and black-box attacks usually rely on a substitute model (Papernot et al., 2017), or gradient estimation of the target model (Ilyas et al., 2018; Uesato et al., 2018).

**Adversarial Robustness.** To mitigate the potential threat of adversarial attacks, extensive research has been conducted, leading to various methods (Guo et al., 2018; Song et al., 2017; Papernot et al., 2016; Madry et al., 2018; Zhang et al., 2019; Atsague et al., 2021). However, some of the proposed methods were later found to be ineffective against strong attacks (Athalye et al., 2018). Adversarial Training (AT) (Goodfellow et al., 2015; Madry et al., 2018), which requires training a classifier with adversarial examples, has been found to be effective to a degree in achieving robustness to adversarial examples. Formally, AT involves crafting adversarial examples during training and solving a saddle point problem formulated as:

$$\min_{\boldsymbol{\theta}} \mathbb{E}_{(\mathbf{x},y)\sim\mathcal{D}} \left[ \max_{\mathbf{x}'\in B_\epsilon(\mathbf{x})} L(f_\theta(\mathbf{x}'), y) \right] \tag{1}$$

where $y$ is the true label of input feature $\mathbf{x}$, $L()$ is the loss function, $\boldsymbol{\theta}$ are the model parameters, and $B_\epsilon(\mathbf{x}) : \{\mathbf{x}' \in \mathcal{X} : \|\mathbf{x}' - \mathbf{x}\|_p \leq \epsilon\}$ represents the $l_p$ norm ball centered around $\mathbf{x}$ constrained by radius $\epsilon$ . In Eq. (1), the inner maximization tries to obtain a worst-case adversarial version of the input $\mathbf{x}$ that increases the loss. The outer minimization then tries to find model parameters that would minimize this worst-case adversarial loss. The relative success of AT (Madry et al., 2018) has inspired various AT variants such as (Zhang et al., 2019; Wang et al., 2019; Ding et al., 2019; Kannan et al., 2018), to cite a few.

**Re-weighting.** Recent works by Zhang et al. (2020) and Liu et al. (2021) have argued for assigning different weights to the losses corresponding to different adversarial examples in the training set. Zhang et al. (2020) assigns weights to an example based on its distance to the decision boundary. Examples that are closer to the decision boundary are assigned larger weights as follows:

$$w(\mathbf{x}_i, y_i) = \frac{1 + tanh(\lambda + 5 \times (1 - 2k(\mathbf{x}_i, y_i)/K))}{2} \tag{2}$$

where $k(\mathbf{x}_i, y_i)$ is the least number of PGD steps to cause misclassification of $\mathbf{x}_i$, $K = 10$ is the number of PGD steps used for generating the attack, and $\lambda$ is set to -1. The assignment function in Eq. (2) is used to reweight the robust losses on each adversarial example in Eq. (1).

Unlike (Zhang et al., 2020) which uses a re-weighting function that is discrete (i.e. the weighting function depends on $k$ PGD iterations) and path-dependent, (Liu et al., 2021) proposed a re-weighting scheme that is continuous and path-independent based on the probability margin between the estimated ground-truth probability of an adversarial example and the probability of the class closest to the ground-truth as follows:

$$PM(\mathbf{x}, y; \theta) = f_\theta(\mathbf{x}')_y - \max_{j, j \neq y} f_\theta(\mathbf{x}')_j \tag{3}$$

The weight assignment function is then defined as:

$$w(\mathbf{x}_i, y_i) = sigmoid(-\gamma(PM_i - \beta)) \tag{4}$$

where $\gamma$ and $\beta$ are hyperparameters.

This work takes a different perspective on reweighting robust losses. We consider the inherent vulnerability of the natural examples used for crafting adversarial examples and the impact of adversarial attacks on each natural example.

## 3  PRELIMINARIES

We use bold letters to denote vectors. We denote $\mathcal{D} = \{\mathbf{x}_i, y_i\}_{i=1}^n$ as a data set of input feature vectors $\mathbf{x}_i \in \mathcal{X} \subseteq \mathbf{R}^d$ and labels $y_i \in \mathcal{Y}$, where $\mathcal{X}$ and $\mathcal{Y}$ represent a feature space and a label set, respectively.

Let $f_\theta : \mathcal{X} \to R^C$ denote a deep neural network (DNN) classifier parameterized by $\theta$ where $C$ represents the number of output classes. For any $\mathbf{x} \in \mathcal{X}$, let the class label predicted by $f_\theta$ be $F_\theta(\mathbf{x}) = \arg\max_k f_\theta(\mathbf{x})_k$, where $f_\theta(\mathbf{x})_k$ denotes the $k$-th component of $f_\theta(\mathbf{x})$. $f_\theta(\mathbf{x})_y$ is the likelihood of $\mathbf{x}$ belonging to class y. $KL(P\|Q)$ is used to represent the Kullback-Leibler divergence of distributions $P$ and $Q$.

We denote $\|\cdot\|_p$ as the $l_p$- norm over $\mathbf{R}^d$, that is, for a vector $\mathbf{x} \in \mathbf{R}^d, \|\mathbf{x}\|_p = (\sum_{i=1}^d |\mathbf{x}_i|^p)^{\frac{1}{p}}$. An $\epsilon$-neighborhood for $\mathbf{x}$ is defined as $B_\epsilon(\mathbf{x}) : \{\mathbf{x}' \in \mathcal{X} : \|\mathbf{x}' - \mathbf{x}\|_p \leq \epsilon\}$. An adversarial example corresponding to a natural input $\mathbf{x}$ is denoted as $\mathbf{x}'$. We refer to a DNN trained only on natural examples as standard network, and the one trained using adversarial examples as robust network or adversarially trained network.

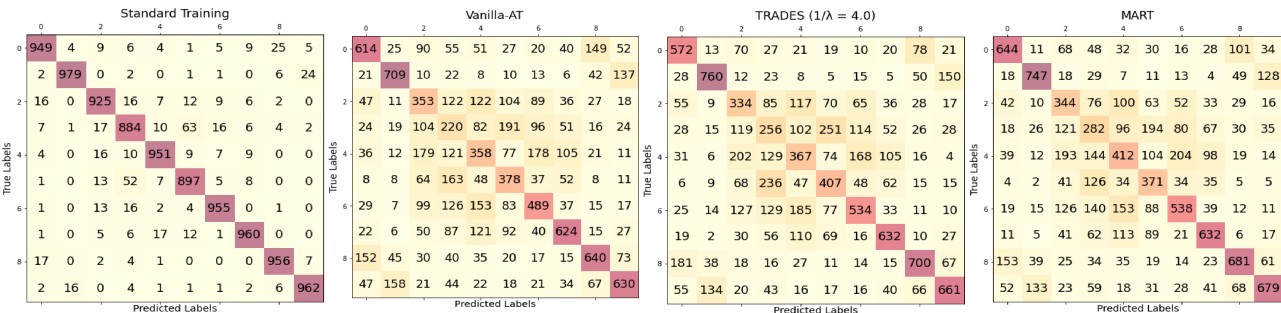

Figure 1: Confusion matrix displaying standard accuracy of ResNet18 on the CIFAR-10 dataset and its robust accuracy under the PGD-100 attack trained by AT variants *vanilla-AT* (Madry et al., 2018), *TRADES* (Zhang et al., 2019), and *MART* (Wang et al., 2019) respectively.

## 4  PROPOSED METHOD

The proposed approach is motivated by two major factors: (1) the unfairness exhibited by adversarial training - adversarial examples crafted from intrinsically vulnerable natural examples are underrepresented in the later stages of adversarial training (Zhang et al., 2020), (2) the discrepancy between individual natural examples and their corresponding attacked variants.

We propose an effective reweighting strategy for assigning importance to robust losses based on these two factors. We will make a case for how the information about the inherent vulnerability of natural examples and the effect of adversarial attacks on these examples may be captured and utilized as inputs to the proposed weight assignment function.

### 4.1  Vulnerability of Natural Examples

Based on the findings in (Xu et al., 2021; Ma et al., 2022), supported by the confusion matrices in Figure 1, it is clear that adversarial training hurts adversarial examples of certain classes more than others, especially adversarial examples crafted from natural examples which are harder to classify under natural training. These natural examples are characterized by their closeness to the class decision boundaries (Xu et al.,

2021; Zhang et al., 2020). Following these findings, we propose a reweighting component to pay attention to adversarial examples whose natural counterparts are more *difficult* to classify.

We argue that a natural example being vulnerable is suggestive of the features of that example correlating to another (wrong) but semantically similar class. Hence, a vulnerable example may be classified with reasonably high confidence to the wrong class. Findings in (Wang, 2021) also indicate that classes that are semantically closer to each other have smaller distances to their decision boundaries; e.g., in the CIFAR-10 dataset, 'cat' is semantically closer to 'dog' than 'airplane,' and 'cat' and 'dog' have smaller distances to the decision boundaries than 'airplane.' In Figure 1, classes '3' ('cat') and '5' ('dog'), which are semantically similar, exhibit more vulnerability under natural training. Input examples corresponding to these classes contain certain similar features. We believe that this relative feature similarity in semantically closer classes makes models less confident in distinguishing their examples. Given this insight, we define the vulnerability of a natural example in terms of a model's estimated class probability.

Given a natural input $\mathbf{x}$ with its corresponding true label $y$, and a model $f_\theta(.)$, the model's estimated probability of $\mathbf{x}$ belonging to $y$ is given as $f_\theta(\mathbf{x})_y$. Note that $y$ is not necessarily the same as $\arg\max_k f_\theta(\mathbf{x})_k$. We formally define the notion of relative vulnerability of inputs below.

**Definition 4.1 (*Relative vulnerability* of examples.)** *Given two input-label pairs $(\boldsymbol{x}_1, y_1)$ and $(\boldsymbol{x}_2, y_2)$, we say the pair $(\boldsymbol{x}_1, y_1)$ is more vulnerable than $(\boldsymbol{x}_2, y_2)$, if $f_\theta(\boldsymbol{x}_1)_{y_1} < f_\theta(\boldsymbol{x}_2)_{y_2}$.*

Based on Definition 4.1, model $f_\theta(.)$ is less confident in estimating the true class of more vulnerable examples. We provide more theoretical explanation to our notion of relative example vulnerability in Appendix A.

Given that adversarial training involves training on adversarial examples, and it unfairly hurts adversarial examples crafted from vulnerable examples, it is intuitive that adversarial training sets up its decision boundary to favor adversarial examples of *invulnerable* classes, as observed in (Xu et al., 2021). This is indicated by the relatively low robust accuracy recorded on certain classes. Given this insight, we consider assigning unequal weights to adversarial examples based on the relative vulnerability of their original (natural) examples with respect to their respective true labels.

Our proposed re-weighting strategy for adversarial training considers the vulnerability of each natural example used to generate adversarial examples for training. This vulnerability for an input $\mathbf{x}$ with label $y$ is given by $f_\theta(\mathbf{x})_y$. Adversarial examples corresponding to natural examples which are intrinsically vulnerable are assigned higher weights. We define a score function denoted as $S_v(\mathbf{x}, y)$ for adaptively assigning importance to input example $\mathbf{x}$ based on a model's confidence that $\mathbf{x}$ belongs to the true label $y$. The score function is as follows:

$$S_v(\mathbf{x}_i, y_i) = \alpha \cdot e^{-\gamma f_\theta(\mathbf{x}_i)_{y_i}} \tag{5}$$

where $\mathbf{x}_i$ is a natural example, $y_i$ is the true label of $\mathbf{x}_i$, $e^{(.)}$ represents an exponential function, and $\gamma >= 1.0$ is a real-valued hyperparameter. $\alpha$ is introduced for numerical stability to ensure that $S_v(\mathbf{x}, y)$ values are not too small and are useful. The formulation in Eq. 5 ensures that higher values are returned for more valnerable examples which have lower $f_\theta(\mathbf{x}_i)_{y_i}$ values. Higher $\gamma$ values will result in a larger disparity in weights between vulnerable examples and less vulnerable examples.

## 4.2 Disparity between Natural and Adversarial Examples

The likelihood estimate of correctly classifying a natural example provides information about its intrinsic vulnerability before an adversarial perturbation is applied to it. However, it does not supply information about the discrepancies in the features learned by the model on natural examples and their corresponding adversarial variants, which largely explain the large disparity between natural and robust errors.

Ilyas et al. (2019) and Tsipras et al. (2018) characterized learned features into robust and non-robust. Robust features refer to features that remain correlated to the true label under adversarial perturbation. In contrast, non-robust features are highly predictive, however, they are brittle and can be anti-correlated with the true label under adversarial perturbations. Tsipras et al. (2018) showed that a DNN classifier is able to learn any useful features on natural examples, but learns only robust features on adversarial examples, assigning zero or infinitesimal weight values to the predictive non-robust features.

The depletion of features in the adversarial examples results in different information loss for different examples. We note that there may be a variation in non-robust features learned by a DNN classifier in the different adversarial examples used in training. For instance, consider two input-label pairs $(\mathbf{x}_1, y_1)$, $(\mathbf{x}_2, y_2)$ and their corresponding perturbed variants $\mathbf{x}'_1$, $\mathbf{x}'_2$. The robust features in $\mathbf{x}'_1$ may have a stronger correlation to $y_1$ than the robust features in $\mathbf{x}'_2$ to $y_2$. In fact, intrinsically vulnerable examples which are relatively weakly correlated with their classes may be affected more by an adversary since their features are further depleted. We propose to score the vulnerability of an example $\mathbf{x}$ as a weight for $\mathbf{x}$ by the discrepancy between the model's output prediction on $\mathbf{x}$ and that on its adversarial variant $\mathbf{x}'$. For simplicity, we score this discrepancies using the KL-divergence as follows:

$$S_d(\mathbf{x}_i, \mathbf{x}'_i) = KL(f_\theta(\mathbf{x}_i) \| f_\theta(\mathbf{x}'_i)) \tag{6}$$

where $S_d$ denotes the proposed discrepancy score function, and $f_\theta(\mathbf{x})$ and $f_\theta(\mathbf{x}')$ denote the model's predictions on natural and the corresponding adversarial examples respectively.

We note that the KL-divergence between a model's predictions on natural examples and adversarial examples is characterized as a boundary error in *TRADES* (Zhang et al., 2019), and is utilized as a regularization term for improving robustness (see Eq. 9). In contrast, here, it is used as a weighting (a value) and the gradient of the KL-divergence in Eq. (6) is not used.

### 4.3 Weight Assignment Function

We propose a weight assignment function for re-weighting robust losses based on the observations in Sections 4.1 and 4.2. The proposed *Vulnerability-aware Instance Reweighting* (*VIR*) function for assigning importance to the example $x_i$ in a training set is as follows:

$$w(\mathbf{x}_i, \mathbf{x}'_i, y_i) = S_v(\mathbf{x}_i, y_i) \cdot S_d(\mathbf{x}_i, \mathbf{x}'_i) + \beta \tag{7}$$

where $\beta$ is a hyperparameter. $S_v(\mathbf{x}, y)$ ensures that emphasis is given to vulnerable examples which are disproportionately hurt by adversarial training. The proposed weight assignment function assigns the largest weights to robust losses corresponding to vulnerable examples having the most significant discrepancy between the model's outputs on the natural and adversarial variants. Furthermore, the least weights are assigned to robust losses corresponding to invulnerable examples having the least discrepancy between the model's outputs on them and their corresponding adversarial variants. The hyperparameter $\beta$ allows us to lower-bound the reweighting function and to balance the relative strength of the lowest and highest weight values.

As a comparison, *GAIRAT* (Zhang et al., 2019) assigns importance to robust losses based on the $k$ PGD steps required to attack an example $\mathbf{x}$. The intuition is that vulnerable inputs take fewer steps to move across the decision boundary. However, as noted in (Liu et al., 2021), this approach is problematic because it is path-dependent, i.e, two inputs with similar starting points may reach their end points using different paths, thus making it unreliable. In addition, the reweighting function is constrained to accepting a few discrete values. In contrast, the proposed reweighting function in eqn (7) takes continuous values and is path-independent.

### 4.4 Applying the Weight Assignment Function

We apply the proposed weight assignment function to prominent adversarial training methods vanilla AT (Madry et al., 2018) and TRADES (Zhang et al., 2019) by assigning importance to robust losses computed by these adversarial training methods. Specifically, we re-write the training objective of the vanilla AT as:

$$\sum_i w(\mathbf{x}_i, \mathbf{x}'_i, y_i) \cdot L_{CE}(f_\theta(\mathbf{x}'_i), y_i) \tag{8}$$

where $L_{CE}$ refers to the cross-entropy loss function. The TRADES robust loss, originally stated as:

$$\sum_i L_{CE}(f_\theta(\mathbf{x}_i), y) + \frac{1}{\lambda} \cdot KL(f_\theta(\mathbf{x}_i) \| f_\theta(\mathbf{x}'_i)), \tag{9}$$

is re-written as:

$$\sum_i L_{CE}(f_\theta(\mathbf{x}_i), y) + \frac{1}{\lambda} \cdot w(\mathbf{x}_i, \mathbf{x}'_i, y_i) \cdot KL(f_\theta(\mathbf{x}_i) \| f_\theta(\mathbf{x}'_i)) \tag{10}$$

where $\lambda$ is a regularization hyper-parameter. We term the training objectives in Eq. (8) and (10) as **VIR-AT** and **VIR-TRADES** respectively.

**Burn-in Period.** During the training, the weight $w(\mathbf{x}_i, \mathbf{x}'_i, y_i)$ is set to 1 in the initial epochs. The application of the proposed weight assignment function is delayed to later epochs. This is because at the initial training phase, the deep model has not sufficiently learned, and thus is less informative. Disregarding this fact may mislead the training process. Zhang et al. (2020) used a similar approach in implementing their re-weighting strategy.

Our proposed VIR-AT algorithm is summarized in the following:

---

**Algorithm 1** VIR-AT Algorithm.

---

**Input:** a neural network model with the parameters $\theta$, step sizes $\kappa_1$ and $\kappa_2$, and a training dataset $\mathcal{D}$ of size n.
**Output:** a robust model with parameters $\theta^*$

1: **for** $epoch = 1$ to num_epochs **do**
2:     **for** $batch = 1$ to num_batchs **do**
3:         sample a mini-batch $\{(x_i, y_i)\}_{i=1}^{M}$ from $\mathcal{D}$;              $\triangleright$ mini-batch of size $M$.
4:         **for** $i = 1$ to M **do**
5:             $\mathbf{x}'_i \leftarrow \mathbf{x}_i + 0.001 \cdot \mathcal{N}(0, 1)$, where $\mathcal{N}(0, I)$ is the Gaussian distribution with zero mean and
6:             identity variance.
7:             **for** $k = 1$ to $K$ **do**
8:                 $\mathbf{x}'_i \leftarrow \prod_{B_\epsilon(\mathbf{x}_i)}(x_i + \kappa_1 \cdot sign(\nabla_{\mathbf{x}'_i} \cdot L(f_\theta(\mathbf{x}'_i), y_i));$     $\triangleright \prod$ is a projection operator.
9:             **end for**
10:             $S_v(\mathbf{x}_i, y_i) \leftarrow \alpha \cdot e^{-\gamma f_\theta(\mathbf{x}_i)_y}$
11:             $S_d(\mathbf{x}'_i, \mathbf{x}_i) \leftarrow KL(f_\theta(\mathbf{x}_i) \| f_\theta(\mathbf{x}'_i))$
12:             $w_i(\mathbf{x}_i, \mathbf{x}'_i, y_i) \leftarrow S_v(\mathbf{x}_i, y_i) \cdot S_d(\mathbf{x}'_i, x_i) + \beta;$     $\triangleright w_i(\mathbf{x}_i, \mathbf{x}'_i, y_i) \leftarrow 1$ if epoch $\leq 76$
13:         **end for**
14:         $\theta \leftarrow \theta - \kappa_2 \nabla_\theta \sum_{i=1}^{M} w_i(\mathbf{x}'_i, \mathbf{x}_i, y_i) \cdot L(f_\theta(\mathbf{x}'_i), y_i)$
15:     **end for**
16: **end for**

---

## 5 EXPERIMENTAL SECTION

In this section, we verify the effectiveness of the proposed re-weighting function through extensive experiments on various datasets including CIFAR-10 (Krizhevsky et al., 2009), CIFAR-100 (Krizhevsky et al., 2009), SVHN(Netzer et al., 2011), and TinyImageNet(Deng et al., 2009). We employed ResNet-18 (RN-18) (He et al., 2016) and WideResNet-34-10 (WRN-34-10) (He et al., 2016) as the backbone models for exploring the effectiveness of the proposed method on CIFAR-10, while CIFAR-100, SVHN, and TinyImageNet are evaluated on ResNet-18.

### 5.1 Experimental Settings

The models are trained for 115 epochs, using mini-batch gradient descent with momentum 0.9, batch size 128, weight decay 3.5e-3 (RN-18) and 7e-4 (WRN-34-10). The learning rates are set to 0.01 and 0.1 for RN-18 and WRN-34-10 respectively. In both cases, the learning rates are decayed by a factor of 10 at 75th, and then at 90th epoch. In *VIR-AT* and *VIR-TRADES*, we introduced the proposed reweighting function on the 76th epoch following (Zhang et al., 2020).

The adversarial examples used during training are obtained by perturbing each image using the Projected Gradient Descent (PGD) (Madry et al., 2018) with the following hyperparameters: $l_\infty$ norm $\epsilon = 8/255$, step-size $\kappa = 2/255$, and $K = 10$ iterations.

## 5.2 Baselines

We compare the robustness obtained using *VIR-AT* and *VIR-TRADES* with prominent AT methods *vanilla-AT* (Madry et al., 2018), *TRADES* (Zhang et al., 2019), and *MART* (Wang et al., 2019). In addition, we compare with the state-of-the-art reweighting schemes *GAIRAT* (Zhang et al., 2020) and *MAIL*(Liu et al., 2021). We conducted additional experiments on recent data augmentation-based defense (Wang et al., 2023) and the results are provided in Appendix B.

## 5.3 Hyperparameters

**Baseline Hyperparameters.** The trade-off hyperparameter $\frac{1}{\lambda}$ is set to 6.0 for training WRN-34-10 and 4.0 for RN-18 with *TRADES*. As recommended by the authors, we set the regularization hyperparameter $\beta$ to 5.0 for training with *MART*.

**VIR Hyperparameters.** The values of constants $\alpha$ and $\beta$ are heuristically determined and set to 7.0 and 0.007 respectively in *VIR-AT* and 8.0 and 1.6 in *VIR-TRADES*. Similarly, we set the value of $\gamma$ to 10.0 and 3.0 in *VIR-AT* and *VIR-TRADES* respectively. We set the value of $\gamma$ to 3.0 for training TinyImageNet with VIR-AT. Also, the value of $\frac{1}{\lambda}$ is set to 5.0 for training *VIR-TRADES*.

## 5.4 Threat Models

The performance of the proposed reweighting function was evaluated using attacks under *White-box* and *Black-box* settings and *Auto attack*.

**White-box attacks.** These attacks have unfettered access to model parameters. To evaluate robustness on CIFAR-10 using RN-18 and WRN34-10, we apply the Fast Gradient Sign Method (FGSM) (Goodfellow et al., 2015) with $\epsilon = 8/255$ ; PGD attack with $\epsilon = 8/255$, step size $\kappa = 1/255$, $K = 100$; CW (CW loss (Carlini & Wagner, 2017) optimized by PGD-20) attack with $\epsilon = 8/255$, step size $1/255$. In addition, we evaluated the robustness of trained models against 100 iterations of $l_\infty$ version of Fast Minimum-norm (FMN) attacks (Pintor et al., 2021). On SVHN and CIFAR-100, we apply PGD attack with $\epsilon = 8/255$, step size $\kappa = 1/255$, $K = 100$. We limited the white-box evaluation on TinyImageNet to PGD-20.

**Black-box attacks.** Under black-box settings, the adversary has no access to the model parameters. We evaluated robust models trained on CIFAR-10 against strong query-based black-box attacks Square (Andriushchenko et al., 2020) with 5,000 queries and SPSA (Uesato et al., 2018) with 100 iterations, perturbation size 0.001 (gradient estimation), learning rate = 0.01, and 256 samples for each gradient estimation. All black-box evaluations are made on trained WRN-34-10.

**Ensemble of Attacks.** Trained models are tested on powerful ensembles of attacks such as *Autoattack* (Croce & Hein, 2020b), which consisting of APGD-CE (Croce & Hein, 2020b), APGD-T (Croce & Hein, 2020b), FAB-T (Croce & Hein, 2020a), and Square (a black-box attack) (Andriushchenko et al., 2020) attacks. In addition, we evaluated the trained models on the Margin Decomposition Ensemble (MDE) attack (Ma et al., 2023).

## 5.5 Performance Evaluation

We summarize our results on CIFAR-10 using RN-18 and WRN-34-10 in Tables 1 and 2, respectively. Moreover, we report results on CIFAR-100, SVHN, and Tiny Imagenet using RN-18 in Tables 3 and 4. Finally, black-box evaluations are made on trained WRN-34-10, and the results are reported in Table 5. Experiments were repeated four times with different random seeds; the mean and standard deviation are subsequently calculated. Results are reported as *mean ± std*.

Table 1: Comparing white-box attack robustness (accuracy %) for ResNet-18 on CIFAR-10. For all methods, distance $\epsilon = 0.031$. We highlight the best-performing method under each attack.

| DEFENSE | NATURAL | FGSM | PGD-100 | CW | FMN-100 | AA | MDE |
|---|---|---|---|---|---|---|---|
| AT | $84.12_{\pm 0.16}$ | $57.88_{\pm 0.13}$ | $51.58_{\pm 0.17}$ | $51.75_{\pm 0.23}$ | $48.92_{\pm 0.25}$ | $47.92_{\pm 0.35}$ | $47.90_{\pm 0.27}$ |
| TRADES | $83.56_{\pm 0.35}$ | $57.82_{\pm 0.32}$ | $52.07_{\pm 0.25}$ | $52.26_{\pm 0.07}$ | $49.74_{\pm 0.30}$ | $48.32_{\pm 0.19}$ | $48.29_{\pm 0.19}$ |
| MART | $80.32_{\pm 0.38}$ | $58.01_{\pm 0.19}$ | $54.03_{\pm 0.28}$ | $49.29_{\pm 0.11}$ | $49.82_{\pm 0.08}$ | $47.61_{\pm 0.27}$ | $47.55_{\pm 0.12}$ |
| GAIRAT | $83.33_{\pm 0.19}$ | $60.20_{\pm 0.29}$ | $54.91_{\pm 0.19}$ | $40.95_{\pm 0.39}$ | $38.62_{\pm 0.29}$ | $32.89_{\pm 0.33}$ | $32.70_{\pm 0.18}$ |
| MAIL-AT | $84.32_{\pm 0.46}$ | $60.11_{\pm 0.39}$ | $55.25_{\pm 0.23}$ | $48.88_{\pm 0.11}$ | $46.83_{\pm 0.17}$ | $44.22_{\pm 0.21}$ | $44.14_{\pm 0.21}$ |
| **VIR-TRADES** | $82.03_{\pm 0.13}$ | $59.62_{\pm 0.08}$ | $54.86_{\pm 0.17}$ | $\mathbf{53.11_{\pm 0.17}}$ | $51.95_{\pm 0.09}$ | $\mathbf{51.03_{\pm 0.16}}$ | $50.89_{\pm 0.12}$ |
| **VIR-AT** | $\mathbf{84.59_{\pm 0.18}}$ | $\mathbf{61.35_{\pm 0.13}}$ | $\mathbf{56.42_{\pm 0.18}}$ | $52.18_{\pm 0.15}$ | $50.56_{\pm 0.12}$ | $48.21_{\pm 0.08}$ | $48.04_{\pm 0.08}$ |

Table 2: Comparing white-box attack robustness (accuracy %) for WideResNet-34-10 on CIFAR-10. For all methods, distance $\epsilon = 0.031$. The best-performing methods under each attack are highlighted.

| DEFENSE | NATURAL | FGSM | PGD-100 | CW | FMN-100 | AA | MDE |
|---|---|---|---|---|---|---|---|
| AT | $86.17_{\pm 0.26}$ | $61.68_{\pm 0.13}$ | $54.45_{\pm 0.31}$ | $55.17_{\pm 0.33}$ | $54.05_{\pm 0.21}$ | $51.90_{\pm 0.28}$ | $51.74_{\pm 0.19}$ |
| TRADES | $85.20_{\pm 0.25}$ | $61.47_{\pm 0.35}$ | $54.81_{\pm 0.31}$ | $56.02_{\pm 0.29}$ | $53.95_{\pm 0.10}$ | $53.09_{\pm 0.18}$ | $52.71_{\pm 0.12}$ |
| MART | $84.59_{\pm 0.11}$ | $62.20_{\pm 0.14}$ | $56.45_{\pm 0.16}$ | $54.52_{\pm 0.11}$ | $53.17_{\pm 0.12}$ | $51.21_{\pm 0.23}$ | $50.92_{\pm 0.15}$ |
| GAIRAT | $85.24_{\pm 0.19}$ | $62.67_{\pm 0.36}$ | $57.09_{\pm 0.27}$ | $44.96_{\pm 0.2}$ | $44.50_{\pm 0.05}$ | $42.29_{\pm 0.11}$ | $41.92_{\pm 0.15}$ |
| MAIL-AT | $84.83_{\pm 0.39}$ | $64.09_{\pm 0.32}$ | $58.86_{\pm 0.25}$ | $51.26_{\pm 0.20}$ | $51.64_{\pm 0.15}$ | $47.10_{\pm 0.22}$ | $47.04_{\pm 0.11}$ |
| **VIR-TRADES** | $84.95_{\pm 0.21}$ | $63.07_{\pm 0.17}$ | $57.56_{\pm 0.21}$ | $\mathbf{56.92_{\pm 0.19}}$ | $\mathbf{55.72_{\pm 0.13}}$ | $\mathbf{54.55_{\pm 0.26}}$ | $54.14_{\pm 0.09}$ |
| **VIR-AT** | $\mathbf{87.13_{\pm 0.36}}$ | $\mathbf{64.71_{\pm 0.29}}$ | $\mathbf{59.82_{\pm 0.29}}$ | $56.11_{\pm 0.16}$ | $54.14_{\pm 0.23}$ | $51.94_{\pm 0.22}$ | $51.83_{\pm 0.12}$ |

Table 3: Comparing white-box attack robustness (accuracy %) for RN-18 on SVHN and CIFAR-100. For all methods, distance $\epsilon = 0.031$.

| | **SVHN** | | | **CIFAR-100** | | |
|---|---|---|---|---|---|---|
| DEFENSE | NATURAL | PGD-100 | AA | NATURAL | PGD-100 | AA |
| AT | $\mathbf{92.94_{\pm 0.46}}$ | $54.74_{\pm 0.28}$ | $45.94_{\pm 0.29}$ | $59.60_{\pm 0.35}$ | $28.57_{\pm 0.25}$ | $24.75_{\pm 0.21}$ |
| TRADES | $92.14_{\pm 0.43}$ | $55.24_{\pm 0.23}$ | $45.64_{\pm 0.29}$ | $60.73_{\pm 0.33}$ | $29.83_{\pm 0.25}$ | $24.83_{\pm 0.29}$ |
| MART | $91.84_{\pm 0.46}$ | $55.54_{\pm 0.21}$ | $43.39_{\pm 0.36}$ | $54.19_{\pm 0.26}$ | $29.94_{\pm 0.21}$ | $25.30_{\pm 0.50}$ |
| GAIRAT | $90.47_{\pm 0.58}$ | $61.37_{\pm 0.28}$ | $37.27_{\pm 0.31}$ | $58.43_{\pm 0.26}$ | $25.74_{\pm 0.41}$ | $17.57_{\pm 0.33}$ |
| MAIL-AT | $91.54_{\pm 0.35}$ | $\mathbf{62.16_{\pm 0.18}}$ | $41.18_{\pm 0.29}$ | $\mathbf{60.74_{\pm 0.15}}$ | $27.62_{\pm 0.0.27}$ | $22.44_{\pm 0.53}$ |
| **VIR-TRADES** | $89.24_{\pm 0.23}$ | $58.63_{\pm 0.32}$ | $\mathbf{50.06_{\pm 0.22}}$ | $59.20_{\pm 0.35}$ | $31.69_{\pm 0.24}$ | $\mathbf{26.25_{\pm 0.25}}$ |
| **VIR-AT** | $91.65_{\pm 0.35}$ | $61.52_{\pm 0.43}$ | $45.91_{\pm 0.41}$ | $59.85_{\pm 0.11}$ | $\mathbf{32.06_{\pm 0.35}}$ | $24.73_{\pm 0.22}$ |

Table 4: Comparing white-box attack robustness (accuracy %) for ResNet-18 on TinyImageNet. Perturbation size $\epsilon = 8/255$ and step size $\kappa = 1/255$ are used for all methods.

| DEFENSE | NATURAL | PGD-20 | AA |
|---|---|---|---|
| AT | $48.79_{\pm 0.15}$ | $23.96_{\pm 0.15}$ | $18.06_{\pm 0.15}$ |
| TRADES | $49.11_{\pm 0.23}$ | $22.89_{\pm 0.26}$ | $16.81_{\pm 0.19}$ |
| MART | $45.91_{\pm 0.24}$ | $26.03_{\pm 0.36}$ | $\mathbf{19.23_{\pm 0.23}}$ |
| GAIRAT | $46.09_{\pm 0.14}$ | $17.21_{\pm 0.33}$ | $12.92_{\pm 0.23}$ |
| MAIL-AT | $49.72_{\pm 0.36}$ | $24.32_{\pm 0.33}$ | $17.61_{\pm 0.35}$ |
| **VIR-TRADES** | $\mathbf{51.17_{\pm 0.19}}$ | $25.82_{\pm 0.13}$ | $18.68_{\pm 0.19}$ |
| **VIR-AT** | $49.09_{\pm 0.25}$ | $\mathbf{26.65_{\pm 0.32}}$ | $18.42_{\pm 0.21}$ |

**Comparison with prominent reweighting methods.** The results in Tables 1-4 show that, in general, the proposed *VIR* method is more effective than the two prominent reweighting methods *MAIL-AT* and

Table 5: Comparing black-box attack robustness (accuracy %) for Wideresnet-34-10 trained on CIFAR-10.

| Defense | Square | SPSA |
|---|---|---|
| AT | $60.12_{\pm 0.25}$ | $61.05_{\pm 0.16}$ |
| TRADES | $59.18_{\pm 0.19}$ | $61.15_{\pm 0.09}$ |
| MART | $58.72_{\pm 0.18}$ | $58.93_{\pm 0.11}$ |
| GAIRAT | $51.97_{\pm 0.10}$ | $52.15_{\pm 0.30}$ |
| MAIL-AT | $58.34_{\pm 0.16}$ | $59.24_{\pm 0.32}$ |
| **VIR-TRADES** | $59.73_{\pm 0.08}$ | $61.45_{\pm 0.26}$ |
| **VIR-AT** | $\mathbf{60.51}_{\pm 0.11}$ | $\mathbf{62.59}_{\pm 0.21}$ |

*GAIRAT*, especially under stronger attacks FMN, CW, and Autoattack. For example, *VIR-AT* significantly outperforms *MAIL-AT* against FMN-100 (+2.5%), CW (+ 5.0%), and Autoattack (+ 5.0%) attacks on CIFAR-10 using WRN-34-10. The proposed *VIR-AT* method achieves large improvement margins over *GAIRAT* against FMN-100 (+ 9.6%), CW (+ 11.1%), and Autoattack (+ 9.6%) on CIFAR-10 using WRN-34-10. *VIR-AT* also consistently performs better than *MAIL-AT* and *GAIRAT* on CIFAR-100 and Tiny-ImageNet against PGD and Autoattack attacks. On SVHN, *MAIL-AT* slightly outperforms our *VIR-AT* method against PGD-100, however, *VIR-AT* performs significantly better against Autoattack by over 4%.

Experimental results presented in Table 5 show that *VIR-AT* performs better than *GAIRAT* and *MAIL-AT* against strong query-based black-box attacks *Square* and *SPSA*. Furthermore, the poorer performance recorded by *GAIRAT* on *black-box* attacks compared to the PGD-100 *white-box* attack may potentially indicate that *GAIRAT* encourages gradient obfuscation according to the arguments made in (Athalye et al., 2018).

**Comparison with non-reweighted variants.** We applied the proposed reweighting method to a prominent variant of vanilla AT, TRADES (Zhang et al., 2019). Experimental results in Tables 1-5 show that *VIR-TRADES* consistently improves upon *TRADES* against all attacks, especially stronger attacks CW, FMN, and Autoattack.

In addition, the confusion matrices displayed in Figure 2 clearly shows the improvement in class-wise accuracies of data-sets corresponding to harder classes. We show in Figure 3 the class-weight distribution (the sum of all sample weights in each class). From Figure 3, we observe that our proposed instance-wise reweighting assigns the largest weight to the most vulnerable class '3' and the least weight to the least vulnerable class '1' for both VIR-AT and VIR-TRADES.

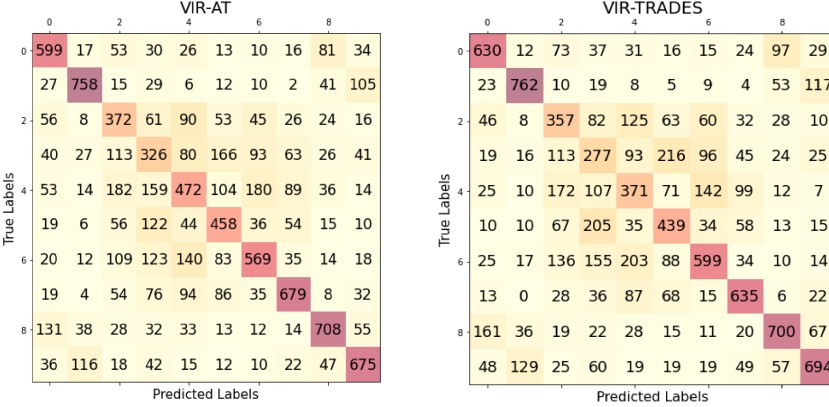

Figure 2: Confusion matrix displaying robust accuracies under PGD-100 attack using VIR-AT and VIR-TRADES for ResNet18 on CIFAR-10 dataset.

Finally, these experimental results show that the existing reweighting methods *GAIRAT* and *MAIL* improved upon the vanilla *AT* against FGSM and PGD attacks but performed much worse against stronger attacks CW, FMN, and Autoattack. In contrast, our proposed *VIR-AT* performed significantly better than *AT*

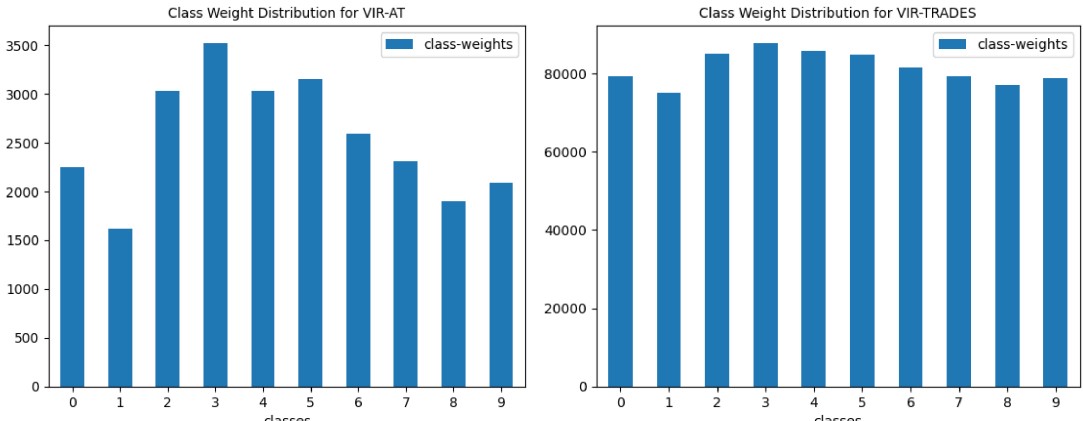

Figure 3: Class-weight distribution (sum of all sample weights in each class) for VIR-AT and VIR-TRADES for ResNet18 during training on CIFAR-10 dataset.

on PGD-100 and FGSM without sacrificing performance against CW, FMN, and Autoattack. A likely explanation for the relatively poor performance of existing reweighting functions on stronger attacks such as Autoattack and CW is that they significantly diminish the influence of less vulnerable examples. While upweighting losses of vulnerable examples can be helpful, over-relying on the vulnerable samples may be detrimental as observed in (Dong et al., 2021). Our proposed VIR function achieved a good balance between weighting vulnerable and less vulnerable examples as shown in the ablation studies presented in the next section.

### 5.6 Ablation Studies and Impact of Hyperparameters

### 5.6.1 Ablation Studies

We conduct ablation studies on the proposed weight assignment function using ResNet-18 on CIFAR-10. The training settings are the same as those described in Section 5.1. We study the influence of each reweighting component on the robustness against white-box and black-box attacks. The results are presented in Table 6.

Table 6: Ablation studies on *VIR-AT* showing the impact of reweighting components proposed in Eq. 5 and 6 on white-box and black-box attack robustness (accuracy %).

| REWEIGHTING COMPONENT | NATURAL | PGD-100 | CW | AA | SQUARE | SPSA |
|---|---|---|---|---|---|---|
| $L_{CE}(f_\theta(x_i'), y_i)$ | $84.12_{\pm 0.16}$ | $51.58_{\pm 0.17}$ | $51.75_{\pm 0.23}$ | $47.92_{\pm 0.35}$ | $55.32_{\pm 0.09}$ | $56.85_{\pm 0.29}$ |
| $S_v(x_i, y_i) \cdot L_{CE}(f_\theta(x_i'), y_i)$ | $84.35_{\pm 0.17}$ | $54.50_{\pm 0.11}$ | $49.61_{\pm 0.19}$ | $45.48_{\pm 0.27}$ | $54.68_{\pm 0.15}$ | $56.08_{\pm 0.25}$ |
| $S_d(x_i, x_i') \cdot L_{CE}(f_\theta(x_i'), y_i)$ | $83.90_{\pm 0.16}$ | $51.17_{\pm 0.14}$ | $51.90_{\pm 0.35}$ | $47.95_{\pm 0.29}$ | $56.22_{\pm 0.19}$ | $56.25_{\pm 0.21}$ |
| $w(x_i, x_i', y_i) \cdot L_{CE}(f_\theta(x_i'), y_i)$ | $\mathbf{84.59_{\pm 0.18}}$ | $\mathbf{56.42_{\pm 0.18}}$ | $\mathbf{52.18_{\pm 0.15}}$ | $\mathbf{48.21_{\pm 0.08}}$ | $\mathbf{56.89_{\pm 0.19}}$ | $\mathbf{57.35_{\pm 0.15}}$ |

Reweighting $L_{CE}(f_\theta(x_i'), y_i)$ with only $S_v(x_i, y_i)$ improves robustness against PGD-100, but yeilds reduced robustness against CW and Autoattack. When $L_{CE}(f_\theta(x_i'), y_i)$ is reweighted using $S_d(x_i, x_i')$, no significant improvement in robustness was observed against PGD-100, however, stable performance over stronger attacks are observed. Reweighting $L_{CE}(f_\theta(x_i'), y_i)$ with the proposed reweighting function significantly improves robustness accuracy against both white-box and black-box attacks.

The ablation studies on *VIR-TRADES* on CIFAR-10 using ResNet-18 are presented in Table 7.

Table 7: Ablation studies on *VIR-TRADES* showing the impact of reweighting components proposed in Eq. 5 and 6 on white-box and black-box attack robustness (accuracy %).

| Reweighting component | Natural | PGD-100 | CW | AA | Square | SPSA |
|---|---|---|---|---|---|---|
| $L_{CE}(f_\theta(x_i), y) + \frac{1}{\lambda} KL(f_\theta(x_i) \| f_\theta(x_i'))$ | $83.56_{\pm0.35}$ | $52.07_{\pm0.25}$ | $52.26_{\pm0.07}$ | $48.32_{\pm0.19}$ | $55.47_{\pm0.13}$ | $56.36_{\pm0.23}$ |
| $L_{CE}(f_\theta(x_i), y) + \frac{1}{\lambda} S_v(x_i, y_i) KL(f_\theta(x_i) \| f_\theta(x_i'))$ | $77.95_{\pm0.11}$ | $54.21_{\pm0.19}$ | $51.70_{\pm0.13}$ | $49.95_{\pm0.15}$ | $55.18_{\pm0.22}$ | $56.11_{\pm0.21}$ |
| $L_{CE}(f_\theta(x_i), y) + \frac{1}{\lambda} S_d(x_i, x_i') KL(f_\theta(x_i) \| f_\theta(x_i'))$ | $\mathbf{86.59}_{\pm0.21}$ | $49.71_{\pm0.18}$ | $49.65_{\pm0.15}$ | $46.77_{\pm0.13}$ | $54.97_{\pm0.11}$ | $55.85_{\pm0.15}$ |
| $L_{CE}(f_\theta(x_i), y) + \frac{1}{\lambda} w(x_i, x_i', y_i) KL(f_\theta(x_i) \| f_\theta(x_i'))$ | $82.03_{\pm-0.13}$ | $\mathbf{54.86}_{\pm0.17}$ | $\mathbf{53.11}_{\pm0.17}$ | $\mathbf{51.03}_{\pm0.16}$ | $\mathbf{56.58}_{\pm0.19}$ | $\mathbf{57.80}_{\pm0.09}$ |

The results in Table 7 show that reweighting *TRADES* with $S_v(x_i, y_i)$ yields better performance than the original *TRADES* against PGD-100 (+ 2.14 %) and Autoattack (+ 1.63 %). However, it yields lower performance against CW (-0.56 %) and natural examples (-5.61%). Reweighting *TRADES* using $S_d(x_i, x_i')$ improves the natural accuracy by +3.03% but yields lower performance against attacks. The proposed reweighting function $w(x_i, x_i', y_i)$ consistently improves *VIR-TRADES* over *TRADES* against PGD-100 (+ 2.97%), CW (+ 0.85%), Autoattack (+ 2.71%), Square attack (+ 1.1%), and SPSA (+ 1.44%). The results show neither $S_v(x_i, y_i)$ nor $S_d(x_i, x_i')$ is sufficient for improving *TRADES* and they achieve the best performance when combined.

### 5.6.2 Impact of Hyperparameters

We provide a brief discussion of the impact of the hyperparameters $\gamma, \alpha$, and $\beta$ used in the proposed reweigthing function. Hyperparameter $\gamma$ appears in the power of the exponential function in Eq. 5. Setting $\gamma$ high results in a large disparity in weights between vulnerable examples and less vulnerable examples. $\alpha$ helps with the numerical stability of Eq. 5. Introducing $\beta$ into the reweighting function in Eq. 7 allows for a balance between relative weights of vulnerable examples and less vulnerable examples. Without adding $\beta$, the reweighting function could potentially return very low values, which significantly diminishes the influence of less vulnerable training samples. On the other hand, if $\beta$ is set too high, the desired reweighting effect is lost. The values of these hyperparameters are heuristically determined. Experimental studies on the impact of the hyperparameters on the performance are shown in Appendix C.

## 6 Conclusion

In this paper, we propose a novel vulnerability-aware instance-wise reweighting strategy for adversarial training. The proposed reweighting strategy takes into consideration the intrinsic vulnerability of natural examples used for crafting adversarial examples during adversarial training. We show that existing reweighting methods fail to achieve significant robustness against stronger white-box and black-box attacks. Lastly, we experimentally show that the proposed reweighting function effectively improves adversarial training without diminishing its performance against stronger attacks. In addition, the proposed method shows improvement on every dataset evaluated.

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

## A  Appendix: Theoretical Explanation of Input Vulnerability

In this section, we show that, under a special setting, the example vulnerability directly corresponds to the probability of predicting the true class . We consider a binary classification task of natural examples sampled from a Gaussian mixture distribution, following the settings described in (Carmon et al., 2019; Schmidt et al., 2018; Xu et al., 2021; Ma et al., 2022). We formally define the settings below.

**Definition A.1 (*Gaussian Mixture Distribution.*)** *Let $-\mu$, $\mu$ be the mean parameters corresponding to data sampled from two classes $y = \{-1, +1\}$ according to Gaussian distribution.*

*Let $\sigma_-, \sigma_+$ represent the variances of the classes -1 and +1 . Then the Gaussian mixture model is defined by the following distribution over $(\mathbf{x}, y) \in R^d \times \{\pm 1\}$:*

$$
y \sim \{-1, +1\} \quad \mu = (\overbrace{\eta, ..., \eta}^{d=dim})
$$

$$
\mathbf{x} \sim \begin{cases} \mathcal{N}(\mu, \sigma_+^2 \mathcal{I}), & \text{if } y = +1 \\ \mathcal{N}(-\mu, \sigma_-^2 \mathcal{I}), & \text{if } y = -1 \end{cases}
\tag{11}
$$

*$\mathcal{I}$ is the d-dimension identity matrix.*

To design classes with different intrinsic vulnerabilities, we ensure that there is a K-factor difference between the variances of the classes such that $\sigma_- : \sigma_+ = 1 : K$ and $K > 1$. The larger variance of the examples corresponding to class +1 intuitively suggests higher vulnerability than an example corresponding to class -1.

The analysis is done using a linear classifier as follows:

$$f(\mathbf{x}) = sign(\langle \omega, \mathbf{x} \rangle + b) \tag{12}$$

where $\omega$ and $b$ respectively represent the model weights and bias term, and $sign(z)$ returns 1 if $z \geq 0$, else $-1$. The natural risk is denoted as:

$$
\begin{aligned}
\mathbf{R}_{nat} &= \mathbb{E}_{(\mathbf{x},y) \sim \mathcal{D}^*}(\mathbb{1}(f(\mathbf{x}) \neq y) \\
&= Pr(f_{nat}(\mathbf{x}) \neq y) \\
&= Pr(y = -1) \cdot Pr(f_{nat}(\mathbf{x}) = +1 | y = -1) + Pr(y = +1) \cdot Pr(f_{nat}(\mathbf{x}) = -1 | y = +1) \\
&= Pr(y = -1) \cdot \mathbf{R}_{nat}^-(f_{nat}) + Pr(y = +1) \cdot \mathbf{R}_{nat}^+(f_{nat})
\end{aligned} \tag{13}
$$

$\mathbf{R}_{nat}^-(f_{nat})$ and $\mathbf{R}_{nat}^+(f_{nat})$ represent the class-wise risk of misclassifying -1 and +1 respectively, $f_{nat}$ is a naturally trained classifier.

**Theorem 1 ((Xu et al., 2021))** *Given a Gaussian distribution $\mathcal{D}^*$, a naturally trained classifier $f_{nat}$ which minimizes the expected natural risk: $f_{nat}(\mathbf{x}) = \arg\min_{f} \mathbb{E}_{(\mathbf{x},y) \sim \mathcal{D}^*}(\mathbb{1}(f(\mathbf{x}) \neq y)$. It has the class-wise natural risk:*

$$\mathbf{R}_{nat}^-(f_{nat}) = Pr\{\mathcal{N}(0,1) \leq A - K \cdot \sqrt{A^2 + q(K)}\}$$

$$\mathbf{R}_{nat}^+(f_{nat}) = Pr\{\mathcal{N}(0,1) \leq -K \cdot A + \sqrt{A^2 + q(K)}\}$$

*where $A = \frac{2}{K^2-1} \frac{\sqrt{d}\eta}{\sigma}$ and $q(K) = \frac{2logK}{K^2-1}$ which is a positive constant and depends only on $K$. Therefore, class +1 has a larger risk: $\mathbf{R}_{nat}^-(f_{nat}) < \mathbf{R}^+(f_{nat})$.*

Theorem 1 shows that the vulnerable class +1 (with larger variance) is more difficult to classify than -1, because the an optimal $f_{nat}$ has a higher standard error for +1 than -1.

**Corollary 1 (Vulnerability of a data sample.)** *The vulnerability of an example may be estimated using a classifier's estimated class-probability.*

**Proof 1** *The class-wise risks corresponding to classes -1 and +1 can be respectively written as:*

$$\mathbf{R}_{nat}^-(f_{nat}) = Pr(f_{nat}(\mathbf{x}) = -1 | y = +1) = Pr(\langle \omega, \mathbf{x} \rangle + b > 0)$$

$$\mathbf{R}_{nat}^+(f_{nat}) = Pr(f_{nat}(\mathbf{x}) = +1 | y = -1) = Pr(\langle \omega, \mathbf{x} \rangle + b < 0)$$

*Let $\mathbf{P}_{nat}^-(f_{nat})$ and $\mathbf{P}_{nat}^+(f_{nat})$ respectively denote the correct probability estimates of classes -1 and +1. Then,*

$$\mathbf{P}_{nat}^-(f_{nat}) = Pr(f_{nat}(\mathbf{x}) = -1 | y = -1)$$

$$\mathbf{P}_{nat}^+(f_{nat}) = Pr(f_{nat}(\mathbf{x}) = +1 | y = +1).$$

*From Theorem 1, $\mathbf{R}_{nat}^-(f_{nat}) < \mathbf{R}_{nat}^+(f_{nat})$. It follows that $\mathbf{P}_{nat}^-(f_{nat}) > \mathbf{P}_{nat}^+(f_{nat})$.*

## B  Additional Experiments on Data Augmentation Based Defense

We perform additional experiments on the data augmentation-based defense using extra datasets generated using the Elucidating Diffusion Model (EDM) following Wang et al. (2023).

We utilize Wideresnet-28-10 as the backbone model for the experiments. The model was trained using 20M EDM-generated data for 400 epochs with the training batch size of 512. Common augmentation described in (Wang et al., 2023) is applied on the training samples. We employ the SGD optimizer with Nesterov momentum (Nesterov, 1983) where the momentum factor and weight decay were set to 0.9 and $5e^{-4}$ respectively. We use the cyclic learning rate schedule with cosine annealing (Smith & Topin, 2019),where the initial learning rate is set to 0.2. For VIR-TRADES, we used the settings described in section 5.3, while for VIR-AT, we set $\gamma$ to 5.0, $\beta$ to 0.2 and $\alpha$ to 7.0.

The obtained results are reported in Table 8. The experimental results show improvement of VIR-TRADES over TRADES and VIR-AT over AT on CIFAR-10

Table 8: Comparing robustness (accuracy %) for Wideresnet-28-10 on CIFAR-10 trained on 20M EDM-generated data.

| Defense | Natural | PGD-100 | CW | AA |
|---|---|---|---|---|
| TRADES | $90.33_{\pm 0.14}$ | $65.74_{\pm 0.16}$ | $64.01_{\pm 0.17}$ | $63.03_{\pm 0.13}$ |
| AT | $91.56_{\pm 0.09}$ | $66.23_{\pm 0.12}$ | $65.62_{\pm 0.09}$ | $62.42_{\pm 0.11}$ |
| **VIR-TRADES** | $89.73_{\pm 0.12}$ | $67.23_{\pm 0.08}$ | $65.47_{\pm 0.05}$ | $\mathbf{64.31}_{\pm 0.17}$ |
| **VIR-AT** | $\mathbf{92.69}_{\pm \mathbf{0.25}}$ | $\mathbf{67.98}_{\pm \mathbf{0.25}}$ | $\mathbf{66.95}_{\pm 0.17}$ | $62.95_{\pm 0.09}$ |

## C  Ablation Studies on Hyperparameters

We show in the following experimental results on varying hyperparameter $\alpha, \gamma, \beta$ values. The results are obtained from training ResNet-18 on CIFAR-10 dataset.

Table 9: Ablation studies on *VIR-AT* showing the impact of the $\beta$ hyperparameter on the performance of the proposed reweighting function.

| Reweighting Function | Natural | PGD-100 | CW | AA |
|---|---|---|---|---|
| $S_v(x_i, y_i) \cdot S_d(x_i, x_i') + (\beta = 0)$ | $84.48_{\pm 0.11}$ | $\mathbf{57.20}_{\pm 0.14}$ | $51.25_{\pm 0.12}$ | $47.32_{\pm 0.11}$ |
| $S_v(x_i, y_i) \cdot S_d(x_i, x_i') + (\beta = \mathbf{0.007})$ | $\mathbf{84.59}_{\pm 0.18}$ | $56.42_{\pm 0.18}$ | $52.18_{\pm 0.15}$ | $\mathbf{48.21}_{\pm \mathbf{0.08}}$ |
| $S_v(x_i, y_i) \cdot S_d(x_i, x_i') + (\beta = 0.1)$ | $84.26_{\pm 0.07}$ | $53.52_{\pm 0.13}$ | $\mathbf{52.20}_{\pm \mathbf{0.19}}$ | $48.17_{\pm 0.13}$ |

Table 10: Ablation studies on *VIR-TRADES* showing the impact of the $\beta$ hyperparameter on the performance of the proposed reweighting function.

| Reweighting Function | Natural | PGD-100 | CW | AA |
|---|---|---|---|---|
| $S_v(x_i, y_i) \cdot S_d(x_i, x_i') + (\beta = 0.5)$ | $\mathbf{83.32}_{\pm 0.18}$ | $53.70_{\pm 0.13}$ | $51.60_{\pm 0.12}$ | $49.29_{\pm 0.09}$ |
| $S_v(x_i, y_i) \cdot S_d(x_i, x_i') + (\beta = \mathbf{1.6})$ | $82.03_{\pm 0.13}$ | $\mathbf{54.86}_{\pm \mathbf{0.17}}$ | $\mathbf{53.11}_{\pm \mathbf{0.17}}$ | $\mathbf{51.03}_{\pm \mathbf{0.16}}$ |
| $S_v(x_i, y_i) \cdot S_d(x_i, x_i') + (\beta = 2.0)$ | $80.86_{\pm 0.10}$ | $54.58_{\pm 0.15}$ | $52.62_{\pm 0.08}$ | $50.56_{\pm 0.13}$ |

Table 11: Ablation studies on *VIR-AT* showing the impact of the $\alpha$ and $\gamma$ hyperparameters on the performance of the proposed reweighting function.

| $\alpha$ | $\gamma$ | $\beta$ | NATURAL | PGD-100 | CW | AA |
|---|---|---|---|---|---|---|
| 1 | 10 | 0.007 | $83.75_{\pm 0.11}$ | $54.38_{\pm 0.08}$ | $52.20_{\pm 0.15}$ | $48.25_{\pm 0.15}$ |
| 2 | 10 | 0.007 | $83.80_{\pm 0.09}$ | $55.28_{\pm 0.05}$ | $52.13_{\pm 0.15}$ | $48.18_{\pm 0.16}$ |
| 3 | 10 | 0.007 | $84.16_{\pm 0.06}$ | $55.85_{\pm 0.10}$ | $52.05_{\pm 0.11}$ | $48.15_{\pm 0.11}$ |
| 5 | 10 | 0.007 | $84.31_{\pm 0.06}$ | $55.92_{\pm 0.10}$ | $51.80_{\pm 0.09}$ | $47.85_{\pm 0.09}$ |
| **7** | **10** | **0.007** | $84.59_{\pm 0.18}$ | $56.42_{\pm 0.18}$ | $52.20_{\pm 0.19}$ | $48.17_{\pm 0.13}$ |
| 8 | 10 | 0.007 | $84.19_{\pm 0.07}$ | $56.68_{\pm 0.12}$ | $52.01_{\pm 0.09}$ | $48.02_{\pm 0.10}$ |
| 7 | 2 | 0.007 | $83.96_{\pm 0.15}$ | $54.75_{\pm 0.13}$ | $51.90_{\pm 0.08}$ | $47.89_{\pm 0.05}$ |
| 7 | 4 | 0.007 | $84.52_{\pm 0.08}$ | $54.92_{\pm 0.10}$ | $52.01_{\pm 0.10}$ | $47.97_{\pm 0.06}$ |
| 7 | 6 | 0.007 | $84.36_{\pm 0.08}$ | $55.82_{\pm 0.11}$ | $51.76_{\pm 0.08}$ | $47.70_{\pm 0.08}$ |
| 7 | 8 | 0.007 | $84.15_{\pm 0.08}$ | $56.30_{\pm 0.11}$ | $51.91_{\pm 0.12}$ | $47.87_{\pm 0.07}$ |

Table 12: Ablation studies on *VIR-TRADES* showing the impact of the $\alpha$ and $\gamma$ hyperparameters on the performance of the proposed reweighting function.

| $\alpha$ | $\gamma$ | $\beta$ | NATURAL | PGD-100 | CW | AA |
|---|---|---|---|---|---|---|
| 1 | 3.0 | 1.6 | $82.10_{\pm 0.15}$ | $54.07_{\pm 0.08}$ | $52.29_{\pm 0.09}$ | $50.08_{\pm 0.12}$ |
| 2 | 3.0 | 1.6 | $82.13_{\pm 13}$ | $54.10_{\pm 0.05}$ | $52.34_{\pm 0.09}$ | $50.13_{\pm 0.08}$ |
| 3 | 3.0 | 1.6 | $82.12_{\pm 0.11}$ | $54.16_{\pm 0.10}$ | $52.51_{\pm 0.10}$ | $50.41_{\pm 0.07}$ |
| 5 | 3.0 | 1.6 | $82.06_{\pm 0.11}$ | $54.31_{\pm 0.08}$ | $52.54_{\pm 0.13}$ | $50.43_{\pm 0.05}$ |
| 7 | 3.0 | 1.6 | $82.05_{\pm 0.10}$ | $54.52_{\pm 0.09}$ | $52.59_{\pm 0.11}$ | $50.52_{\pm 0.05}$ |
| **8** | **3.0** | **1.6** | $82.03_{\pm 0.13}$ | $54.86_{\pm 0.17}$ | $53.11_{\pm 0.17}$ | $51.03_{\pm 0.16}$ |
| 8 | 1.0 | 1.6 | $81.05_{\pm 0.10}$ | $53.96_{\pm 0.13}$ | $51.85_{\pm 0.06}$ | $49.89_{\pm 0.09}$ |
| 8 | 2.0 | 1.6 | $81.09_{\pm 0.12}$ | $54.26_{\pm 0.15}$ | $52.41_{\pm 0.08}$ | $50.28_{\pm 0.17}$ |
| 8 | 4.0 | 1.6 | $81.91_{\pm 0.09}$ | $54.42_{\pm 0.13}$ | $52.70_{\pm 0.07}$ | $50.34_{\pm 0.11}$ |

