# OpenReview forum: "Vulnerability-Aware Instance Reweighting For Adversarial Training"
_TMLR — Accepted by TMLR_

### Review · Reviewer_YFuN · 2023-05-30

**Summary Of Contributions:**

This paper studies the imbalance problem in adversarial training and proposes to reweight the importance of each training sample based on the prediction disparity between the nature and adversarial version of the model. The assumption is that a large disparity indicates more vulnerability, which is proved from the perspective of risk-to-class-probability with Gaussian distribution. The proposed reweighting scheme demonstrates improvement in standard adversarial training and TRADES.

Overall, the key contribution is the use of prediction change to reweight a training sample during adversarial training.

**Audience:**

Yes

**Broader Impact Concerns:**

No obvious concerns have been identified.

**Claims And Evidence:**

Yes

**Requested Changes:**

1. Show the class weight distribution (sum over all sample weights in the class)  caused by the reweighting during training.
2. Test against the MDE attack.
3. Show its compatibility with recent SOTA defense, i.e., the data augmentation based defense.

**Strengths And Weaknesses:**

Strengths:
1. The study of a challenging problem in adversarial machine learning, i.e.,  how to train adversarially robust models.
2. The proposed idea is simple enough to be applied to any existing AT method.
3. The improvement to TRADES according to AutoAttack seems to be significant.

Weaknesses:
1. Not sure if instance reweighting is better than class reweighting or class-class reweighting. Certain classes are known to be easily attacked into other classes. How the proposed approach can address this type of imbalance?
2. Should the instance reweighting consider class imbalance problem, as some classes are notably harder than others? How the class-wsie accumulative of the instance reweighting aligns with the class distribution?
3. It is not clear if the proposed method is compatible with recent data augmentation methods, e.g., the one [1] on the robustbench leaderboard.
4. Since the reweighting will skew the logits distribution, the author should also test against the MDE attack [2] to prove it does not cause imbalanced gradients.

[1] Better Diffusion Models Further Improve Adversarial Training.
[2] https://github.com/HanxunH/MDAttack

---

### Review · Reviewer_bYBD · 2023-06-04

**Summary Of Contributions:**

This paper proposes a new strategy for weighting instances during adversarial training. In the traditional adversarial training method (without weighting), depending on the original class label, adversarial examples have different effects on the resulting robust accuracy. Some prior works tried to reduce this unfairness among adversarial examples by re-weighting the training loss and the authors propose another weighting strategy.

The authors’ weighting is based on the vulnerability of the natural sample $\mathbf{x}$ and the discrepancy between the model predictions $f_\theta(\mathbf{x}_i)$ and $f_\theta(\mathbf{x}’_i)$. First, the vulnerability factor $S_v$ is measured by the model’s confidence $f_\theta(\mathbf{x})_y$ in the event that natural sample $\mathbf{x}$ belongs to class $y$. Then, the discrepancy factor $S_d$ is measured by the KL divergence of the model prediction probabilities $f_\theta(\mathbf{x})$ on a natural example $\mathbf{x}$ from the model prediction probabilities $f_\theta(\mathbf{x}’)$  on the adversarial example $\mathbf{x}’$ generated from $\mathbf{x}$. Then the weighting $w$ is determined as a linear function $w=\alpha S_v \cdot S_d + \beta$ for predetermined parameter $\alpha$ and $\beta$. Finally, the authors propose the use of this weighting with some burn-in period so that the model can learn enough information about the data.

An ample amount of experiments are performed to present the improvement from the proposed weighting method. For most experiments, the adversarial training with the proposed method demonstrated descent robust accuracy compared to the existing methods. Especially, the proposed method successfully improved the robust accuracy against some attacks, e.g., CW and AutoAttack, whereas the existing method failed to maintain the robust accuracies from the adversarial training. The authors also present some ablation studies to show the effects of each weighting factor.


**Audience:**

Yes

**Broader Impact Concerns:**

I don’t see a particular broader impact concern regarding this paper.

**Claims And Evidence:**

Yes

**Requested Changes:**

1. I can see some writing issues, e.g., grammar, notations, etc. I suggest the authors read through the writing.
    1. I can see many citations in parentheses in the middle of a sentence, and this sometimes makes the writing hard to read. Please distinguish the use of `\citet{}` (citation as a part of a sentence) and `\citep{}`(citation not a part of a sentence).
    2. The notations are not used consistently. For example, the class label $y$ is sometimes written in boldface in cases where it is definitely not a vector. Even boldface letters appear in different typefaces. Please go through the doc and make the notations consistent.
2. In my opinion, the authors should emphasize the failures of existing weighting strategies against CW and AutoAttack more.
    1. In particular, ablation studies show that the introduction of $S_d$ was crucial for the robustness against those attacks. Can the authors relate this to the reason why the prior methods fail against the stronger attacks?
3. While determining the hyperparameters heuristically makes sense to me, it would be better to have some experimental study on these hyperparameters.
    1. For example, my understanding of $\beta$ is that it lower bounds the weighting so that the training loss (in Eq. (11)) does not disappear for very low $S_v$ and $S_d$. Based on this understanding, setting $\beta=0.007$ seems quite small to me and I want to see the effect of setting this parameter higher.
4. One minor request: I don’t think that Section 4.2 has more valuable information than the ablation study in Appendix. (Of course, it is a meaningful theoretical insight, but this paper is not a theory-intensive work, anyway.) Consider moving Section 4.2 to Appendix and moving the ablation study in Appendix to the main body of the paper.


**Strengths And Weaknesses:**

# Strength

1. Experiments are done extensively on different models and different datasets against various attacks. Also, the results support the improvement well, demonstrating the success of the authors reweighting method.
    1. The proposed method shows good performance against stronger attacks, e.g., CW and AutoAttack, whereas the existing methods failed to maintain their performances.
2. Ablation studies demonstrate the effect of each weighting factor well. The observations on each factor would be useful for other researchers who study the reweighting in AT.

# Weaknesses
1. I can observe that $S_v$ (weighting factor from the vulnerability) is bounded, but $S_d$ (weighting factor from the KL divergence) is not bounded. This sounds like that, for some adversarial examples, the weighting could be mainly dominated by $S_d$.
    1. Can the authors provide the distribution of $S_v$ values and $S_d$ values during the training? Experimental evidence that $S_d$ does not explode dramatically would be enough to support the authors' strategy.
    2. If it is not possible to provide such evidence, isn’t it better to bound $S_d$ somehow?

---

### Review · Reviewer_m7G5 · 2023-06-11

**Summary Of Contributions:**

This paper investigates on a better instance reweighting method to help metigate the discrepancy of robust accuracy between easy and hard samples. Based on the observations and shortcomings of previous work, this work proposes a vulnerability measurement unitlizing true class probablity and disparity between natural and adversarial output. Experiments are provided to show the effectiveness of the proposed method on improving robust accuracy, and ablation study is conducted on the proposed vulnerablity metric.

**Audience:**

Yes

**Broader Impact Concerns:**

No concern on broader impact

**Claims And Evidence:**

Yes

**Requested Changes:**

Please discuss more on the impact of hyperparameters (e.g. $\alpha, \gamma$ in Equ. (5) and $\beta$ in Equ. (10)) on the performance of the proposed method

**Strengths And Weaknesses:**

## Strength

This paper is overall well written and well motivated. The paper provides clear discussion on why the reweighting is needed, how the proposed reweighting criteria is proposed, and how is it different from previous methods. The formulation of the porposed method makes sense, and emperical results are promixing on multiple models and datasets.

## Weakness

The proposed method invloves multiple hyperparameters in the formulation of the reweighting criteria, yet their usage and imact is not well discussed. More analysis and ablation study should be provided on how these hyperparameters impact the performance, and how should they be chosen in practice.

---

### Decision · Action_Editors · 2023-07-11

**Recommendation:** Accept as is

**Comment:**

The authors have successfully addressed some major comments raised by the Reviewers during phase-1 review.  As a result, all reviewers are satisfied with the revision and agree that the paper is interesting for audience and provide sufficient evidence to support their claims.

**Audience:**

The topic of adversarial training is interesting for TMLR's audience.  The findings of this paper will add certain values to this field.

**Claims And Evidence:**

This paper proposes a new strategy for weighting instances during adversarial training. The key contribution is the use of prediction change to reweight a training sample during adversarial training. The experimental evaluations and comparisons are convincing to support their claims.